# Carbon Hybrid Materials—Design, Manufacturing, and Applications

**DOI:** 10.3390/nano13030431

**Published:** 2023-01-20

**Authors:** Anuptha Pujari, Devika Chauhan, Megha Chitranshi, Ronald Hudepohl, Ashley Kubley, Vesselin Shanov, Mark Schulz

**Affiliations:** 1College of Engineering and Applied Sciences, University of Cincinnati, Cincinnati, OH 45221, USA; 2College of Design, Art, Architecture, and Planning, University of Cincinnati, Cincinnati, OH 45221, USA

**Keywords:** carbon nanotubes, FC-CVD, composite, air filtration, bioaerosol filtration, carbon fiber, nanoparticles, nanomaterials

## Abstract

Carbon nanotubes (CNTs) have extraordinary properties and are used for applications in various fields of engineering and research. Due to their unique combination of properties, such as good electrical and thermal conductivity and mechanical strength, there is an increasing demand to produce CNTs with enhanced and customized properties. CNTs are produced using different synthesis methods and have extraordinary properties individually at the nanotube scale. However, it is challenging to achieve these properties when CNTs are used to form macroscopic sheets, tapes, and yarns. To further improve the properties of macroscale forms of CNTs, various types of nanoparticles and microfibers can be integrated into the CNT materials. The nanoparticles and microfibers can be chosen to selectively enhance the properties of CNT materials at the macroscopic level. In this paper, we propose a technique to manufacture carbon hybrid materials (CHMs) by combining CNT non-woven fabric (in the form of sheets or tapes) with microfibers to form CNT-CF hybrid materials with new/improved properties. CHMs are formed by integrating or adding nanoparticles, microparticles, or fibers into the CNT sheet. The additive materials can be incorporated into the synthesis process from the inlet or the outlet of the reactor system. This paper focuses on CHMs produced using the gas phase pyrolysis method with microparticles/fibers integrated at the outlet of the reactor and continuous microfiber tapes integrated into the CNT sheet at the outlet using a tape feeding machine. After synthesis, characterizations such as microscopy and thermogravimetric analysis were used to study the morphology and composition of the CNTs, and examples for potential applications are discussed in this paper.

## 1. Introduction to Carbon Nanotube/Carbon Fiber (CNT-CF) Hybrid Materials

Carbon nanotubes are manufactured in various methods; however, chemical vapor deposition (CVD) is the most common way of synthesizing CNTs [1,2]. The floating catalyst CVD method can produce CNTs > 1 mm long. The CNTs produced by this method are usually intertangled in a web wrapped onto a drum to produce CNT sheets. The macroscale CNT sheets have versatile properties and are a good candidate for forming hybrid materials [3,4,5,6,7,8,9,10]. In this work, we will examine the integration of carbon fibers in the intertangled web of CNTs synthesized by the floating catalyst chemical vapor deposition method in a single-step process. The synthesized carbon nanotube/carbon fiber (CNT-CF) hybrid material can be used to fabricate laminated composites with improved properties. In addition, chopped carbon fibers and carbon fiber tows are integrated into the CNT sheets.

Carbon fibers (CF) are continuous microfibers about 7 microns in diameter with high strength, high elastic modulus, and low density and are used as reinforcement material in many composite material applications. Continuous CF can be aligned and densely packed in a composite material. These fibers can also be woven or braided to form the fabric infiltrated with resin and cured to form composites. Carbon fiber laminated composites have high strength, electrical and thermal conductivity, and a high elastic modulus along the fiber axis compared to the transverse direction [11,12]. Due to their lightweight and high strength, carbon fiber laminated composites are widely used in various structural applications.

Additionally, non-woven CF can also be formed into a sheet by compression. The versatile properties of these laminated composites have attracted huge continuous research attention for applications in various fields [13]. Here, carbon fiber reinforcement can be used in different forms, including mesh, tows, and chopped fibers, and the epoxy polymer acts as a matrix material in laminated composites [14]. The epoxy polymer enables laminating the carbon fiber fabrics by working as a binder to achieve the required thickness. The polymer also provides stiffness for structural applications. The primary role of the polymer matrix is to transfer stresses between fibers. The matrix offers support under compression loading by preventing fiber buckling, providing a barrier to the environment, and protecting the fiber’s surface from mechanical abrasion. However, the low fracture toughness of the brittle matrix resin system makes laminated composites prone to crack initiation and propagation. The most common failure mechanisms in laminate composites originate in the resin matrix, such as matrix cracking, fiber-matrix interface debonding, and interlaminar delamination [15].

Along with these failure modes, the lack of fiber reinforcement in the through-the-thickness direction makes laminated composites susceptible to impact damage and crack from bearing loads. Additionally, the low glass transition temperature of the resin matrix makes the laminated polymer matrix composite unsuitable for high-temperature applications. Because of these reasons, a high fiber volume fraction (V_f_) is desirable to improve the composite’s mechanical and physical properties [16,17].

In order to reduce the problem of multiple failure modes in composites, different forms of carbon nanotubes are incorporated into laminated composites to improve the composite’s resistance to crack initiation and propagation. Integration of CNTs also improves the electrical conductivity and heat resistance properties of fiber-reinforced polymer composites (FRPC). Carbon nanotube powders have traditionally been used as nanofillers to improve the fracture toughness and electrical conductivity of laminated composites [18,19,20].

However, the improvement obtained when CNT powder is used as a nanofiller is limited as it is not a continuous reinforcement (matrix separates the nanotubes) [21]. To further enhance the properties of laminated composites, CNT sheets can be interleaved to improve the in-plane strength and interlaminar toughness and reduce interlaminar delamination in fiber-reinforced laminated composites [3,4,5,19,21,22,23,24,25]. In another approach to enhance laminated composites’ tensile strength and interfacial shear strength, carbon nanotubes are grafted on the carbon fiber surface to provide reinforcement in the out-of-plane direction [17,26].

However, integrating CNT sheets in the laminated composite process is tedious while following the traditional methods of composites synthesis. Additionally, the lack of proper resin infiltration of the CNTs in laminated composites can result in defect sites [27,28,29]. Here, we will examine a single-step method for in situ integration of CNTs in FRPC to produce CNT-CF hybrids.

## 2. Carbon Hybrid Material (CHM) Manufacturing

This section generally describes the methods used to manufacture CHM. The problem is that CNT materials such as fabric, yarn, and tapes are expensive, and their properties need to be improved to open up new applications. Likewise, the properties of laminated composite materials also need to be improved to open up more applications, especially where high toughness and high conductivity are essential [30]. One possible solution that we propose is to integrate (i) nanoparticles (NPs) (metals, ceramics, and other carbon materials) and (ii) microfibers (carbon fiber, Kevlar, chopped and continuous fibers) into CNT fabric to form hybrid materials with new/improved properties and lower cost than CNT materials alone. A modified gas phase pyrolysis (floating catalyst) method is used to perform the synthesis of CNT fabric. The synthesis for the horizontal floating catalyst method has been described in our previous article [2,31]. The fuel mixture consists of a 9:1 methanol (Sigma Aldrich, St. Louis, MO, USA, 99.8%) and hexane (Fisher Scientific, Waltham, MA, USA, 95%) mixture, 1% ferrocene (Sigma Aldrich, St. Louis, MO, USA, 98%), and 0.3% thiophene (Sigma Aldrich, St. Louis, MO, USA, 99%) mixed via a sonicator. The fuel is injected into a two-inch OD alumina horizontal reactor tube at high temperatures. The reactor system is purged with Ar gas, and the reactor tube is heated to 1420 °C throughout the synthesis process. The purging process includes six gas exchanges to ensure a minimal O_2_ concentration in the reactor. The liquid fuel injected evaporates and decomposes into catalyst particles and a carbon source. The CNTs are grown from the catalyst, which aggregates to form a long CNT sock. The sock is collected at the outlet of the ceramic tube and is rolled into a sheet with the help of a translator covered with a Teflon sheet wound on a drum in a glovebox [32,33]. Video presentations that further describe the methods are given in [34].

### 2.1. Carbon Hybrid Materials (CHM) Manufacturing System and Example Materials Produced

A synthesis was performed in a reactor consisting of an inlet for fuel and particle injectors, and an outlet harvest box with functions that include densification of the sheet, optional particle incorporation, winding, tape laying, and tape or sheet coating. The custom fuel injector was developed to atomize and inject the fuel. In addition, a custom functional NP injector was designed to inject dry nanoparticles (NPs) into the inlet of the reactor. Figure 1 shows an image of the custom fuel injector used for injecting dry nanoparticles. A graphical representation of the synthesis process is shown in Figure 2. The feedstock is injected at the inlet as a fine mist with help of an atomizer. This then decomposes into nano-sized catalyst particles onto to which the CNTs nucleate and form a CNT sock which is then collected at the outlet on winding drum in a glove box.

A machine for wrapping carbon fiber (CF) onto CNT fabric was developed. The tape-laying machine is used at the outlet of the synthesis reactor and is a continuous process; see Figure 3. Integrating CF tape into the CNT fabric increases the fabric’s strength. As a result, rolls of CNT-CF tape can be commercialized. Figure 4 shows the tape coating/layering machine to coat CF tape with CNT. Each CNT (about 1 mm long) crosses about 70 carbon fibers, increasing the CF material’s strength. Coating the CNT sock onto CF is a continuous process, but the spool of CF tape must be exchanged. If desired, the CF tape can be coated on two sides in two rounds.

The CF tape can also be layered onto CNT fabric directly from the synthesis reactor. Rolls of CNT-CF tape can be commercially produced. Coating CNT onto CF spread tow tape (30 gsm) may provide new higher-strength composites.

Nanoparticles can also be integrated into the CNT fabric, as shown in Figure 5. Again, phase diagrams are used to guide the reaction. For example, Zn melts at 420 °C and then boils or vaporizes at about 907 °C, releasing Zn atoms into the reactor. It is assumed that the Zn atoms agglomerate and form spheres and are then deposited onto the CNT sock, similar to ferrocene releasing Fe atoms that agglomerate. It is unknown exactly how the different nanoparticles and ferrocene mixtures affect the synthesis reaction. 

A large number of combinations of compound nanoparticles (formed by ball-milling different materials together) can be used in the nanotube reactor. Some different carbons shapes, e.g., cones and spirals, have been developed (Figure 6). The particles integrated into the CNT fabric also affect the properties of the fabric in different ways that are being explored.

### 2.2. Carbon Hybrid Material Design

The carbon hybrid material (CHM), in general, is composed of: (i) a CNT fabric matrix; (ii) carbon fiber (CF) reinforcing material, (iii) metal nanoparticles (NPs), and (iv) a thin polymer binder. A unit cell of the CHM, shown in Figure 7, is 10 microns wide. The scale of the components of CHM spans three orders of magnitude: CNTs (circa 7 nm); CNT strands (7–70 nm); nanoparticles (70–700 nm); and microfibers (7000 nm). Imagination can be used to design new materials by choosing the components.

### 2.3. Model of the In-Plane Tensile Strength (Transverse to the Fiber Direction) for CHM Sheet

A model for CHM fabric is shown in Figure 8. In the model, carbon fiber (CF) (grey) is contained in epoxy (orange) reinforced with CNT fabric (red). Figure 8 shows how failure can occur in a conventional composite. Figure 9 shows how failure can happen in a CNT-reinforced composite [35].

The tensile strength of the composite is predicted using the rule of mixtures (ROM) and must be verified by testing. Figure 10 of CHMs is used to visualize how more force is required to pull CNTs (red) over carbon fibers. Compressive and bending strength must also be verified by testing. The expected volume fractions of materials are given in Table 1, Table 2 shows the strength of material components, and Table 3 presents the strength of the composites.

Based on the results in Table 3, the predicted increase in tensile strength of CFRP 0°/90° = ((1.71 − 1.45)/1.45) (100) = 18%. Strength increases as the volume fraction of CNT fabric increases. The prediction must be verified by testing. Although the increase in strength is modest, it may be important for aerospace and other lightweight applications [36,37].

### 2.4. A Family of Carbon Hybrid Materials (CHM)

A variety of CHM materials can be formed; see Figure 11, Figure 12, Figure 13 and Figure 14. These are initial samples. Further optimization of the process and the characterization of material properties must be performed. Figure 11 shows the fabric used for filtering. Figure 12 shows a cylinder and rings. Figure 13 fronts a strip. Figure 14 shows a sheet.

The CFRP-CNT material is a cross-between the properties of CNT fabric and CF fabric. As a result, the material is expected to have a higher volume fraction of fiber than conventional composites.

CNT fabric about 10 microns thick or thicker is almost impermeable to air and water. To make the fabric breathable, thin layers are used, as shown in Figure 14. CNT bundles in the fabric are 10–70 nanometers in diameter. These strands are like a string; they do not have bending stiffness and drape and fill in the fabric’s porosity. This is unlike rigid microfibers that have bending stiffness, produce pores, and are breathable in thicker sections. Thus, designing CNT filter membranes is more complicated than designing filters using microfibers. The nanoscale size of the bundles may provide filtering with a smaller pressure drop or filtering of smaller-size particles. Figure 15 shows a partly transparent CNT sheet sample that can be used for potential filtering applications.

## 3. Initial Characterization of CNT-CF Hybrid Materials

CNT-CF sheet is initially characterized using microscopy and thermogravimetric analysis.

### 3.1. Microscopic Analysis of CNT-CF Sheet

Scanning electron microscopic (SEM) characterization of the CNT-CF hybrid fabric gives the material’s microscopic surface morphological information. The SEM characterization was performed at the University of Cincinnati Advanced Materials Characterization Center using an FEI Apreo SEM. Figure 16 shows the microscopic image of the chopped carbon fiber integrated into the CNT web. Similarly, we employed a single-step in situ process to interleave CNT sheets for FRPC by sandwiching carbon fiber tows between CNT layers, as shown in Figure 17.

Figure 17 shows the SEM image of the carbon fiber tows sandwiched between CNT sheets, which can be directly used to manufacture CFPC plies interleaved with CNT sheets by hand lay-up and resin infusion methods. The intertangled web of the CNTs firmly integrates the carbon fiber tows to resist the separation of CNT sheets from the carbon fibers. In addition, the porosity in the CNT sheets allows good resin infiltration and can resist interlaminar delamination.

### 3.2. Thermogravimetric Analysis

Carbon nanotube integration is vital for improving the thermal stability of CFRC. Therefore, the thermal degradation of the chopped carbon fiber and chopped carbon fiber integrated into the CNT matrix is analyzed with the help of a Netzsch STA409 thermal gravimetric analysis (TGA) machine. For TG analysis, CNT sheets of weights between 3 mg and 4.5 mg are heated up from room temperature to 1300 °C at a rate of 5 °C min^−1^ in 100 mL min^−1^ of air. Figure 18 shows the TGA graph for chopped carbon fiber and chopped carbon fiber integrated into the CNT matrix.

Under oxidizing conditions, chopped carbon fiber and CNT carbon fiber’s thermal degradation starts at ~600 °C. At 780 °C, the chopped carbon fiber completely decomposes without leaving any residue, whereas the CNT chopped carbon fiber leaves a minimal amount of metal residue. The metal residue is from the iron catalyst used for CNT synthesis. It is seen that the carbon fiber integrated CNTs hybrid material has similar thermal stability as carbon fiber. Therefore, reducing the matrix volume fraction and increasing the filler volume fraction increases the laminated composites’ operating temperature. This research also aims to replace the polymer matrix with the CNT fabric. This polymer-free composite would be a fabric material with high strength. The first derivate of the TG graph gives us information on the change in weight % with respect to the change in temperature in °C, shown in Figure 19.

The first derivative of the TGA graph shows that the weight change with respect to temperature is greater for chopped carbon fiber than CNT. This implies that integrating the CNTs in the carbon fiber improves the chopped carbon fiber’s thermal characteristics.

## 4. Application Examples

Generally, the major applications of hybrid fabric fall into textile manufacturing uses, as illustrated in Figure 20. CHM fabric is composited (plied) with conventional fabric to improve the properties of the textile. Textiles with CHM inside can be used in various applications, from personal protective equipment to polymeric composite materials. The use of CHM in textile engineering has been discussed in our previous work [6,7,8,9]. Here, we discuss the use of CHMs for filtering applications.

The porous structure of CNTs membranes leads to various applications in molecules’ transport, separation, and filtration processes. The carbon nanotubes’ internal diameters are in various bacterial and virus pathogens, making them a promising candidate for filtering applications. The CNT sheet is porous but breathable only in very thin sheets. Thicker CNT sheets are nearly impermeable. The diameter of the CNTs can be controlled by controlling the size of the catalyst particle [38]. Small iron nanoclusters produce small single-walled carbon nanotubes (SWCNTs) with a few double-walled carbon nanotubes (DWCNTs), medium-sized catalysts produce SWCNTs and thin-walled multi-walled carbon nanotubes (MWCNTs), and large-diameter catalyst particles produce MWCNTs only. The role of sulfur is also associated with nanotube surface chemistry and growth. The amount of sulfur in the feedstock can also produce single-, double-, and multi-walled carbon nanotubes. Other factors such as synthesis temperature, the flow rate of carrier gases, hydrocarbon type, and catalyst also play an important role in carbon nanotube synthesis [31,39]. Functionalization of CNTs can make strong interfacial bonds between nanotubes and hence improve the material’s mechanical strength. Functionalization also solves the lack of solubility issue [40]. The porosity of the CNT surface can be increased by oxidation treatment. The use of nitric acid and other oxidation agents with high oxygen content increases the strength of the CNT sheet due to the hydrogen bonding interactions [41].

### 4.1. Air Filtration

The filtration efficiency and pressure drop are important parameters for evaluating the filter performance. A good filter needs high filter efficiency and a low-pressure drop. Large specific areas of CNTs allow more aerosols and CNT surface interactions than any material. The critical challenge for the CNT filter is its small pore size, which can easily clog due to the collection of particles on the surface, resulting in a higher pressure drop. When this pressure drop value reaches a specific value, these filters need to be discarded, limiting filter life [38,42]. Coated filters have been developed to overcome the pressure drop problem without compromising the permeability of the filter. The CNT-metal filter in which carbon nanotubes were grown directly upon the micron-sized metallic fibers of the filter using thermal chemical vapor deposition showed higher efficiency without a significant change in the pressure drop [43]. A novel filter was developed with an aligned CNT sheet and embedding the CNT sheet in between polypropylene melt-blown non-woven fabrics. The filtration efficiency was tested on multiple CNT sheets in parallel and cross-ply configurations. The three-layer CNTs cross-ply configuration had the highest quality factor and met the high-efficiency particulate air (HEPA) filter criterion. The increase in quality factor might be due to the cross-ply structure of CNTs that produced an equivalent effect of decreasing the average orientation of the CNTs. Aligned carbon nanotube membranes were produced perpendicular on the micromachined Si/Sio_2_ wafer for air filtering. The permeability of the carbon nanotube membranes was controlled by the growth time of the nanotubes using the chemical vapor deposition (CVD) process. Membranes with a short growth time showed maximum permeability due to the strong flow in the partially filled channels. High-density packed nanometer-pore-sized nanotubes in microporous and mesoporous structures offered efficient filtering of sub-micron-sized particles [44].

### 4.2. Bioaerosol Filtration

A study of narrow-diameter SWCNTs demonstrated severe damage and cell inactivation when they are in direct cell contact with SWCNTs. The damage to the bacteria is caused by physical damage to the outer membrane, which causes the release of intercellular content [45]. An SWCNT-MWCNT hybrid filter was designed to overcome the limitations of the individual filter. MWCNTs were selected for the filter matrix due to their high permeability, low cost, and suitable adsorption property, whereas SWCNTs were deposited on the filter matrix to make a thin upper layer to take advantage of the antibacterial property of SWCNTs. The hybrid filter showed high virus removal compared to SWCNTs and MWCNTs filters [46]. CNT filters produced with different pore sizes and CNT loadings showed that the removal efficiency increased with increased CNT loading, irrespective of the membrane support and pore sizes. Acid treatment of CNT filters doubled the removal efficiency because of the removal of metal impurities from the surface, which led to a stronger adhesion ability of CNT [47]. MWCNT-deposited glass-fiber filters were prepared using spark discharge and chemical vapor deposition methods. The particle filtration efficiency increased for nanoparticles and submicron particles without affecting the pressure drop across the glass-fiber filter [48]. A CNT-based facepiece respirator was designed, and a manikin-based system was used to evaluate its filtration efficiency. Three types of filters with different SWCNTs loading on polypropylene filters were prepared. The filtration efficiency of the CNTs filters was improved with the increased concentration of the SWCNT [49].

As discussed above, integrating CNTs into textiles opens a wide area of applications [6]. Therefore, a detailed study of the integration of CNTs into textiles for wearable technology was performed [8]. Carbon nanotubes can produce a fabric that can filter air, water, and other contaminants and exhibit good strength, current carrying capability, and light weight.

To benefit the small diameter, high surface area, porous structure of CNTs, and antibacterial properties of silver nanoparticles, a CNT silver nanofilter was designed to remove viruses and pathogens from the water. The nanofilter was plasma treated to create a hydrophilic surface. The nano-sized pore filter successfully removed the micro-sized bacteria from the water [50]. CNTs membranes for water filtering applications have been discussed in our previous paper [24].

### 4.3. Zn-CNT Battery

Carbon zinc batteries were the first commercial batteries and are low cost and produce a voltage of about 1.5 V. Alkaline and lithium batteries have a longer battery life than carbon zinc batteries but are more expensive. The Zn-C battery works based on an electrochemical reaction between zinc and manganese dioxide enabled by an electrolyte. They are constructed using a Zn can, the anode, and the battery’s negative electrode. They can contain a layer of NH_4_Cl or ZnCl_2_ aqueous paste [51,52,53,54] impregnating a paper layer that separates the Zn can from a mixture of powdered carbon and MnO_2_. A carbon rod cathode or positive electrode is located in the center of the battery. The carbon rod electrode does not corrode in the salt-based electrolyte.

A thin battery is formed using the CNT fabric and Zn sheet in a saltwater solution; see Figure 20. The Zn-CNT battery is based on galvanic potential [54] and produces a 1 V open circuit in the dilute electrolyte. Thin batteries can be joined in series to increase the voltage. The Zn dissolves and forms a white milky fluid. When the Zn is mostly dissolved, the CNT film can be removed and placed into a new Zn multi-plate housing, making the battery partially reusable, instead of discarding the battery. The CNT film is the most expensive component of the battery. Millions of pounds of Zn-C batteries are discarded each year around the world [51,52,53,54]. The proposed Zn-CNT reconstructable battery reduces the environmental impact of batteries. Another option for the use of CNT is the dual-carbon electrode battery market which is expected to record a CAGR of 10% during 2020–2025. Dual-carbon batteries have both an anode and cathode made of carbon. Fast charging, with low manufacturing cost compared to other batteries and eco-friendly with recyclable features, they are expected to drive the dual-carbon battery market. However, the high cost of CNT and the competition from other battery technologies hinder CNT adoption for batteries. CNT hybrid materials can provide customized properties for batteries that may reduce size and cost.

## 5. Conclusions

This paper discusses the individual properties of CNTs and CFRCs. An in situ synthesis technique for integrating CNTs with CFs was also discussed. Integrating particles and fibers into CNT fabric formed different carbon hybrid materials. Hybrid carbon nanotube/carbon fiber (CNT-CF) was manufactured as stated above. The CHMs contained CNTs and CFs, a non-woven nanofabric comprised of CF sandwiched and intertwined between nanotube webs. Many combinations of nanoparticles and microfibers can be used to design different CHM. The advantages of forming CNT/microfiber composites are to improve modestly many of the properties of laminated composites.

## Figures and Tables

**Figure 1 nanomaterials-13-00431-f001:**
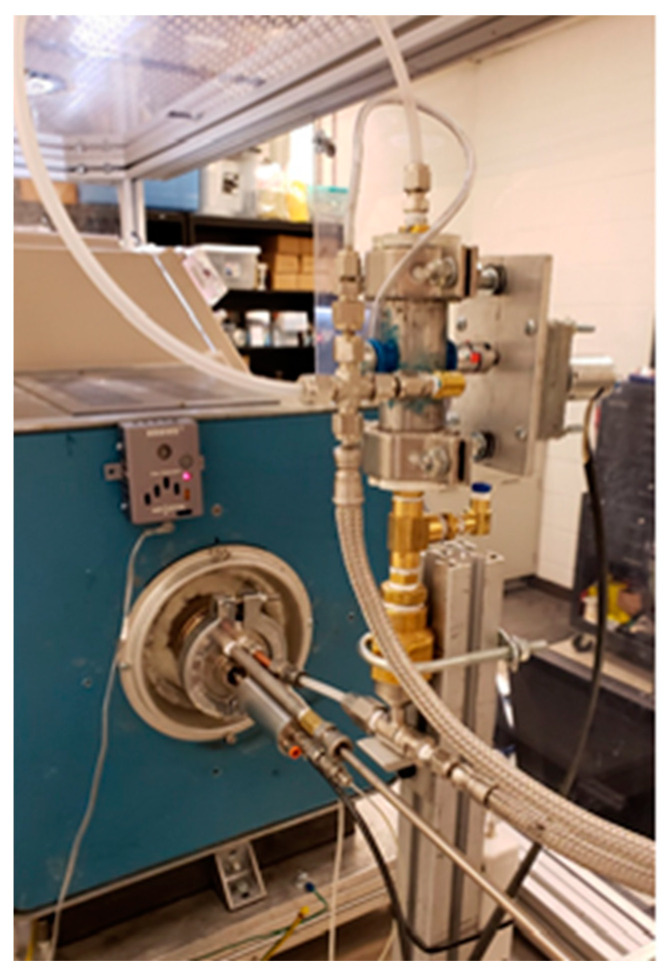
Custom fuel injector and particle injector attached to the inlet of the reactor.

**Figure 2 nanomaterials-13-00431-f002:**
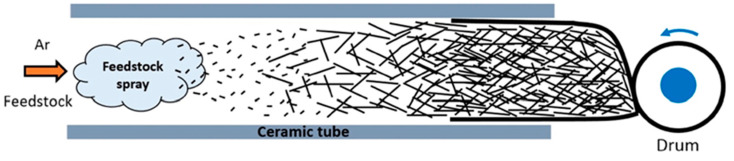
Schematic representation of gas phase pyrolysis (floating catalyst) method used for synthesis.

**Figure 3 nanomaterials-13-00431-f003:**
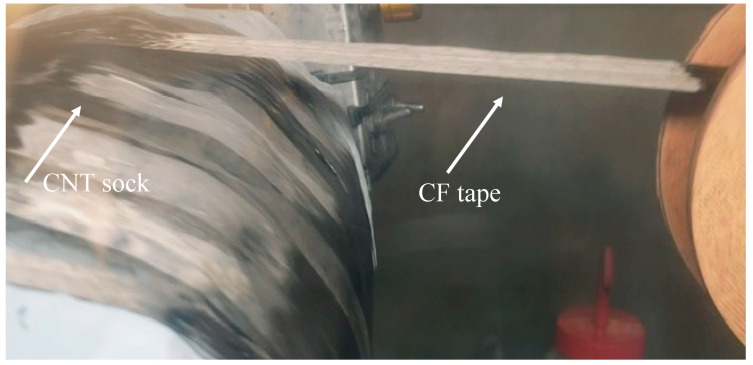
Winding drum used at the outlet for simultaneous wrapping of CNT and carbon fiber (CF) spread tow high modulus (HM) tape to make strong and stiff CHM.

**Figure 4 nanomaterials-13-00431-f004:**
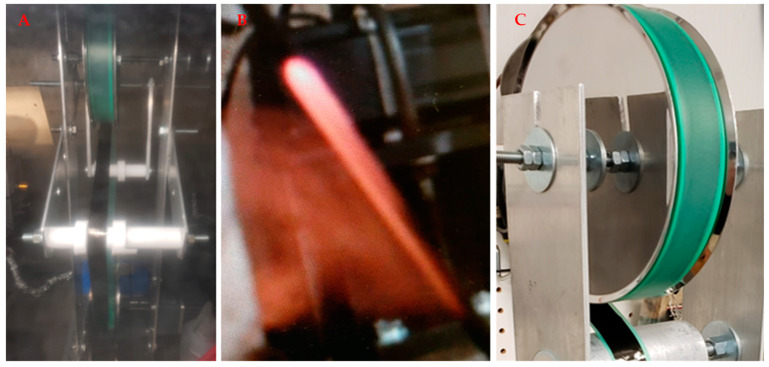
Tape coating and layering machine used for wrapping of CF tape onto CNT. (**A**) CNT can be coated onto the tape. (**B**) The red is the reflection of the light caused due to hot sock at the outlet from the reaction process. (**C**) The CNT web or sock exits the reactor tube and is deposited onto the CF tape and wound from one spool to another in the tape-laying machine.

**Figure 5 nanomaterials-13-00431-f005:**
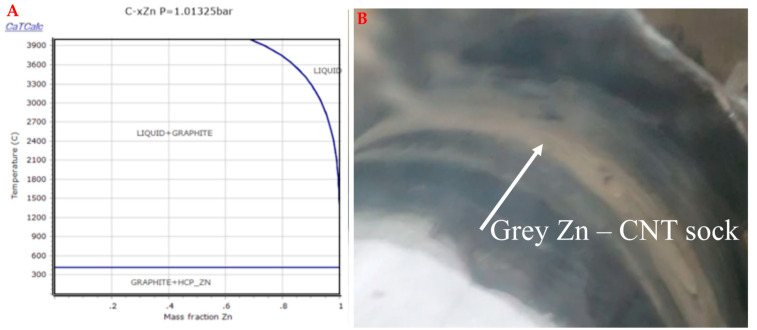
Incorporating nanoparticles into the synthesis process. (**A**) Phase diagram to describe mixing Zn with C. Zn NPs (60 nm diameter) melt, vaporize, and are deposited on the CNT sheet or inside the CNT. (**B**) CNT sock winding onto the take-up drum to form a CNT sheet with Zn NPs only in the center (grey-colored strip). The nanoparticles can be functionally graded throughout the CNT fabric.

**Figure 6 nanomaterials-13-00431-f006:**
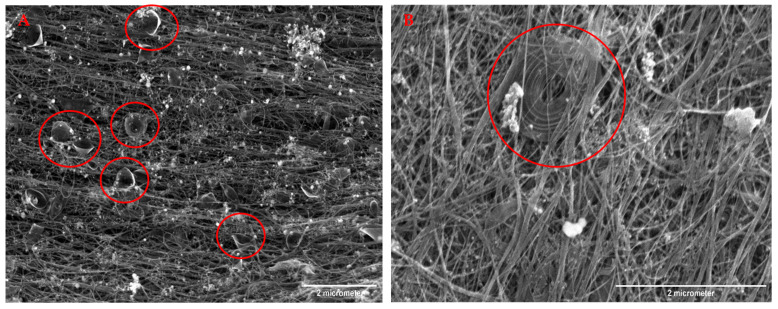
SEM images of CNT cones and spirals. (**A**) CNT cones in red circles. (**B**) Spiral CNT in the red circle.

**Figure 7 nanomaterials-13-00431-f007:**
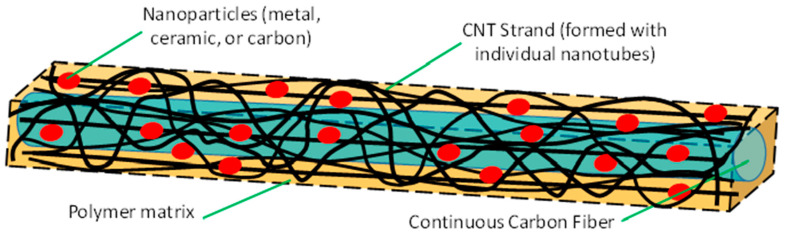
Schematic of an idealized unit cell of a carbon hybrid material.

**Figure 8 nanomaterials-13-00431-f008:**
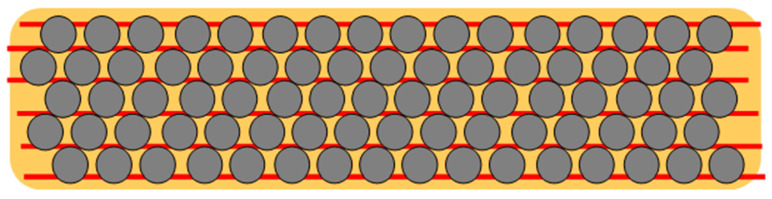
Model for CHM sheet. Illustrated is carbon fiber (CF) (grey) in epoxy (orange) reinforced with CNT fabric (red).

**Figure 9 nanomaterials-13-00431-f009:**
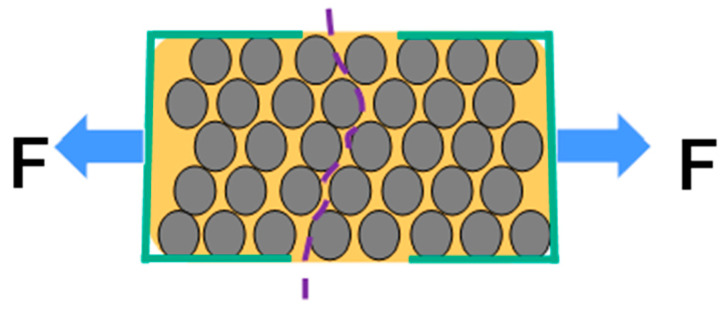
Illustration of a conventional unidirectional ply. The epoxy cracks at a low load (purple dash line).

**Figure 10 nanomaterials-13-00431-f010:**
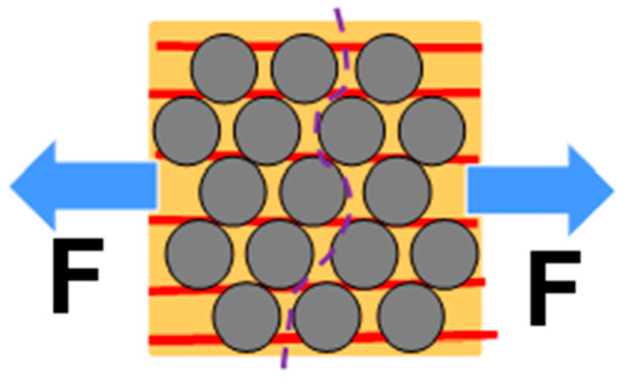
Illustration of CHM unidirectional ply. The CNT strands must be pulled out over many CFs to cause failure in the direction transverse to the fiber.

**Figure 11 nanomaterials-13-00431-f011:**
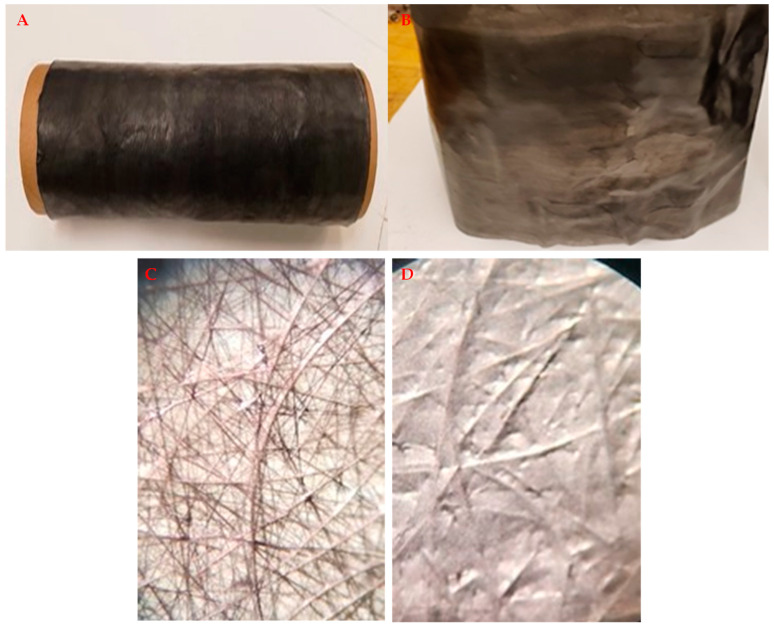
Tissue/CNT fabric. (**A**,**B**) Formed by layering thin CNT onto carbon fiber tissue. A drum of tissue-CNT is opened and wrapped on a spool. This material is used for filtering applications. (**C**) CNT filter fabric, showing the tissue side, where tissue/CNT is partly transparent. (**D**) Showing the CNT side for the tissue/CNT fabric.

**Figure 12 nanomaterials-13-00431-f012:**
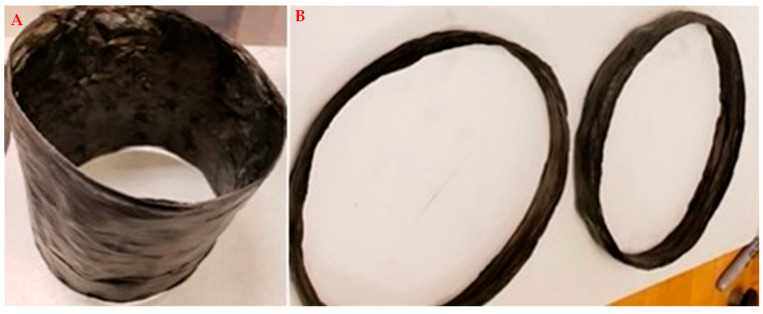
CF-CNT cylinders (**A**,**B**) rings. The material is flexible (not as brittle as CFRP) and is expected to have good tensile strength and toughness.

**Figure 13 nanomaterials-13-00431-f013:**
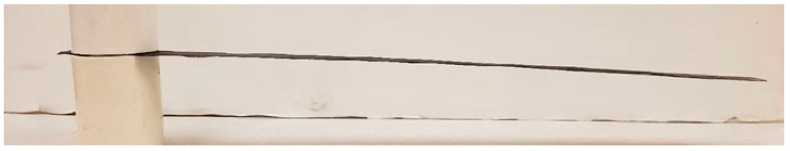
CFRP-CNT strip (cantilevered, with thin polymer binder). The material is lower density than CFRP, more flexible, and is expected to have lower strength in compression than CFRP.

**Figure 14 nanomaterials-13-00431-f014:**
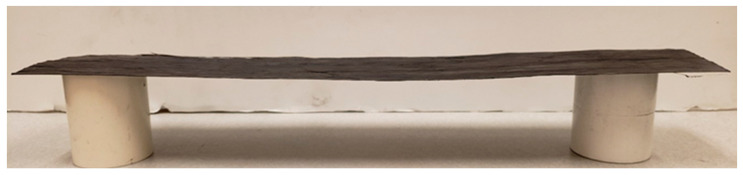
CFRP-CNT sheet (end supported, with thin polymer binder).

**Figure 15 nanomaterials-13-00431-f015:**
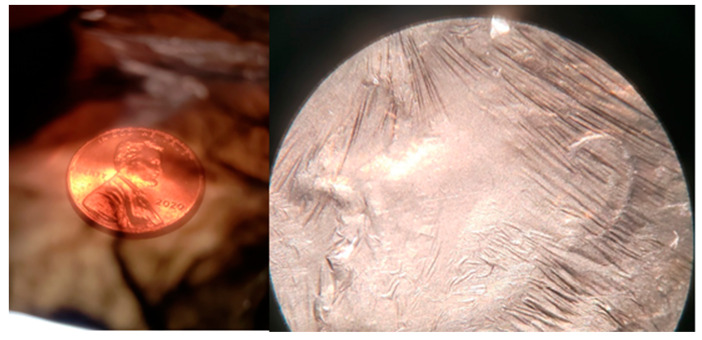
CNT film 2–3 microns thick is partly transparent, showing a US penny coin. CNT film covering a US dime coin.

**Figure 16 nanomaterials-13-00431-f016:**
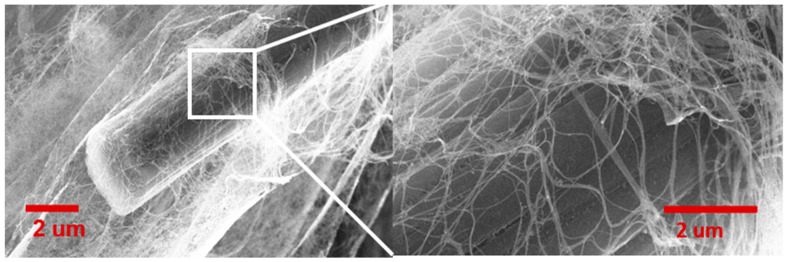
SEM image of chopped carbon fiber integrated into the intertangled web of CNT fabric.

**Figure 17 nanomaterials-13-00431-f017:**
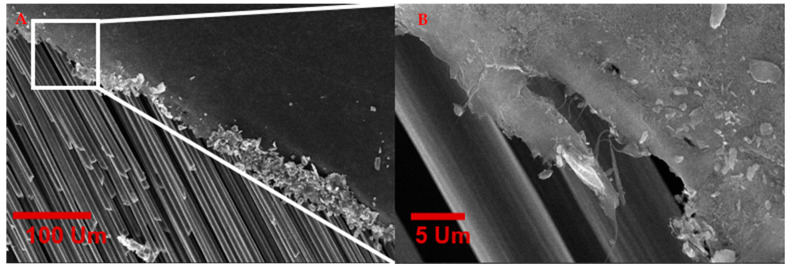
(**A**,**B**). SEM images of carbon fiber tows sandwich between CNT sheets produced from floating catalyst CVD method in a single-step integration method at two different magnifications.

**Figure 18 nanomaterials-13-00431-f018:**
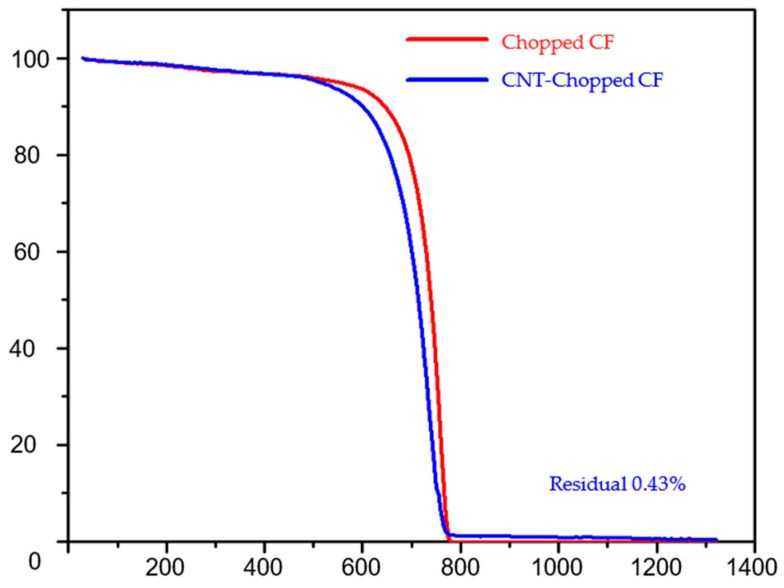
Thermogravimetric analysis graph for chopped carbon fiber and CNT-integrated chopped fiber (there is no polymer).

**Figure 19 nanomaterials-13-00431-f019:**
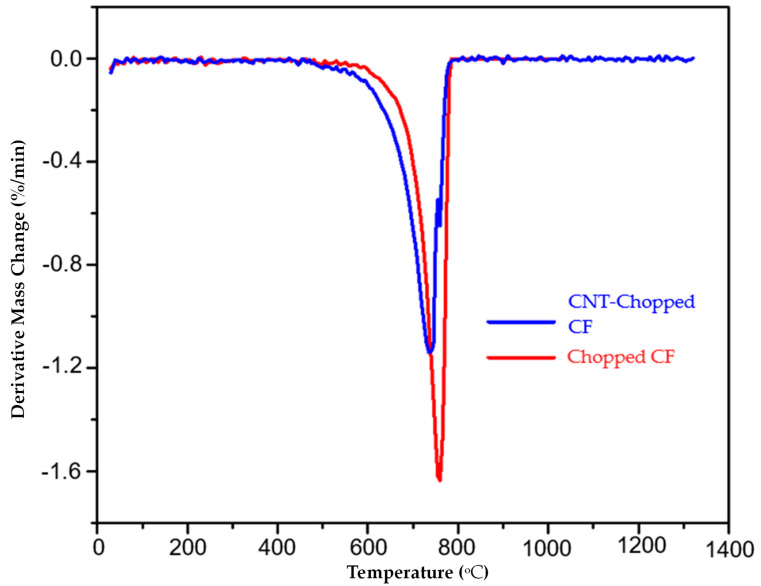
The first derivative of the TG graph for chopped carbon fiber and CNT-integrated chopped fiber.

**Figure 20 nanomaterials-13-00431-f020:**
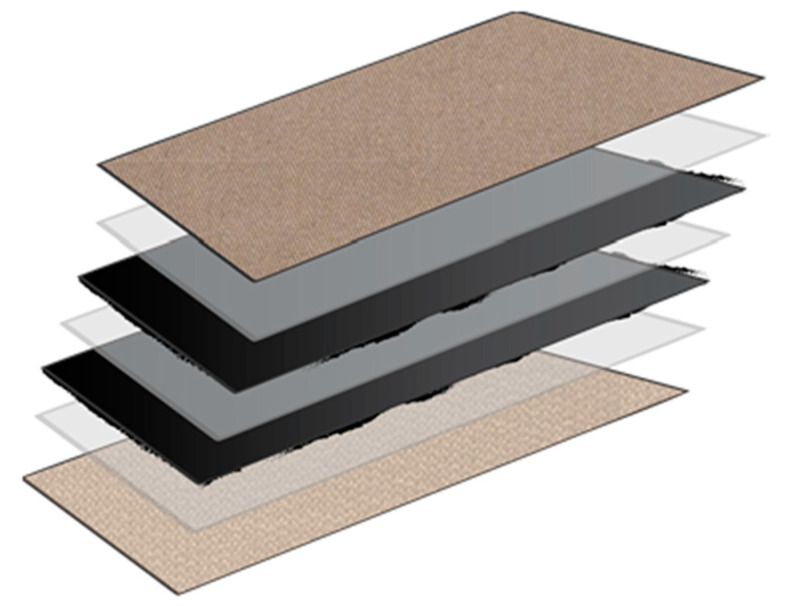
Composited textile with CHM layers (black) inside (figure by Ashley Kubley).

**Table 1 nanomaterials-13-00431-t001:** Expected volume fractions of components in a CNT-CF composite.

Composite Material	V_f_ Epoxy	V_f_ CF	V_f_ CNT Fiber
CFRP (unidirectional)	0.3	0.7	0.0
CFRP-CNT (unidirectional)	0.2	0.7	0.1

**Table 2 nanomaterials-13-00431-t002:** Strength of material components.

Material Components	Epoxy	HM CF	CNT Fibric Macroscale	CNT Fibric Microscale
Strength (GPa)	0.07	4.0	0.4	5.0

**Table 3 nanomaterials-13-00431-t003:** Rule of mixtures strength of composites.

Composite Material	Strength 0° (GPa)	Strength 90° (GPa)
CFRP 0°	4(0.7) + 0.07(0.3) = 2.82	0.07(1) = 0.07
CFRP 0°/90°	(2.82 + 0.0)/2 = 1.45	1.45
CFRP-CNT 0°	4(0.7) + 0.07(0.2) + 0.4(0.1) = 2.85	0.7(0.9) + 5(0.1) = 0.56
CFRP-CNT 0°/90°	(2.85 + 0.56)/2	1.71

## Data Availability

The data presented in this study are available on request from the corresponding author.

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
