# Peer review of "Carbon Hybrid Materials—Design, Manufacturing, and Applications"

_nanomaterials, 2023, doi:10.3390/nano13030431_

Round 1

Reviewer 1 Report

The paper entitled “Carbon Hybrid Materials – Design, Manufacturing, and Applications” presents the design, manufacturing, and applications of CNT materials reinforced by nanoparticles and microfibers. Specifically, Carbon Hybrid Materials (CHMs) were prepared by combining CNT non-woven fabric (in the form of sheets or tapes) with microfibers. This paper focuses on CHMs produced using the gas phase pyrolysis method with microparticles/fibers integrated at the outlet of the reactor and continuous microfibers tapes integrated into the CNT sheet at the outlet using a tape feeding machine. Then, CHMs were characterized and their application fields were discussed. The research content of the design, manufacturing, and applications of CHMs is in the scope of Nanomaterials and interesting. However, after reading the paper, I have following comments/suggestions, which require the author to make major revisions seriously in the manuscript.

1. In "Abstract", the research background and preparation methods of CHMs are mainly introduced, but the performance characterization and application results of CHMs are not summarized.

2. In " Keywords", the number of keywords exceeds ten, and the number of periodicals is required to be three to ten.

3. In "1 Introduction", the author needs to provide the basis for the experiment on why to choose the carbon fiber with a diameter of 7 microns.

4. In the full text, the sentences written by the author are in the present tense without any change in the tense. It is suggested that the past tense be used in some results description and literature review.

5. At the same time, many sentences use the active voice, and it is recommended to use the passive voice. For example, "Also, the layer of proper resin injection of the CNTs in laminated composites can result in defect sites [27 – 29]."

6. The pictures in Figures 1 to 5 are not marked with words. In order to ensure the smooth reading of readers, they need to be marked in the pictures. What’s more, Figure 4 and Figure 5 are not clear and need to be replaced with higher resolution images

7. Figures 8 to 10 are not marked, and the schematic diagram is very simple. The author needs to beautify the schematic diagram to enrich the picture.

8. The processing and testing equipment used in the paper need to be provided the manufacturer, model and other information, the author needs to make additional explanations.

9. Figure 19 only has the title but no picture, so the author needs to add. At the same time, the author gives a simple description of the prepared CHMs properties, which cannot be regarded as performance characterization. The author needs further analysis.

10. The serial number of "Summary and Conclusions" in the paper is incorrect, and the author needs to correct it.

11. In "4 Summary and Conclusions ", it is suggested that the author should discuss the conclusions separately so that the research results can be read more clearly and easily. Such as "(1) ..…. (2) ……"

Author Response

The responses have an attached. 

-------------------------------------

  1. In "Abstract", the research background and preparation methods of CHMs are mainly introduced, but the performance characterization and application results of CHMs are not summarized.

Response: Thank you for the comment, the abstract has now been updated with an introduction to characterizations and applications used and discussed in this paper

  1. In " Keywords", the number of keywords exceeds ten, and the number of periodicals is required to be three to ten.

Response: Thank you for the comment, the keywords section has now been updated

  1. In "1 Introduction", the author needs to provide the basis for the experiment on why to choose the carbon fiber with a diameter of 7 microns.

Response: We thank the reviewer for the worthy notice. Carbon Hybrid Materials (CHM) manufacturing is accomplished by incorporating micron size carbon fibers in nano-size CNT matrix. Therefore, for better integration of the two materials in the presented work, we have selected the smallest diameter size of the carbon fibers available on shelf. Incorporating larger diameter carbon fiber can introduce defects from voids in the carbon hybrid material. However, integrating larger diameter carbon fiber in the CNT matrix and its resulting effect on the material properties of carbon hybrid material will be studied in future.

  1. In the full text, the sentences written by the author are in the present tense without any change in the tense. It is suggested that the past tense be used in some results description and literature review.

Response: We thank the reviewer for the useful suggestions. The manuscript has been revised appropriately.

  1. At the same time, many sentences use the active voice, and it is recommended to use the passive voice. For example, "Also, the layer of proper resin injection of the CNTs in laminated composites can result in defect sites [27 – 29]."

Response: We thank the reviewer for the useful suggestions. The manuscript has been revised appropriately.

  1. The pictures in Figures 1 to 5 are not marked with words. In order to ensure the smooth reading of readers, they need to be marked in the pictures. What’s more, Figure 4 and Figure 5 are not clear and need to be replaced with higher resolution images

Response: We thank the reviewer for the useful suggestions. The manuscript has been revised appropriately.

  1. Figures 8 to 10 are not marked, and the schematic diagram is very simple. The author needs to beautify the schematic diagram to enrich the picture.

Response: We thank the reviewer for the useful suggestions. The simple schematics are prepared to easy perception of the novel idea this paper is pitching. To maintain the simplicity of the schematic, the caption of the schematics provides more information about the schematic.

  1. The processing and testing equipment used in the paper need to be provided the manufacturer, model and other information, the author needs to make additional explanations.

Response: We thank the reviewer for the useful suggestions. The manuscript has been revised appropriately.

  1. Figure 19 only has the title but no picture, so the author needs to add. At the same time, the author gives a simple description of the prepared CHMs properties, which cannot be regarded as performance characterization. The author needs further analysis.

Response: Thank you for the comment, figure 19 has now be updated. This paper mostly focuses on the synthesis of CHMs, and we are continuing to work on the characterizations as part of our next paper. We are also submitting a beamline proposal to SLAC – SSRL for performing WAXD on our samples.

  1. The serial number of "Summary and Conclusions" in the paper is incorrect, and the author needs to correct it.

Response: The serial number has now been corrected. Thank you for the comment.

  1. In "4 Summary and Conclusions ", it is suggested that the author should discuss the conclusions separately so that the research results can be read more clearly and easily. Such as "(1) ..…. (2) ……"

Response: We thank the reviewer for the useful suggestions. We have now updated the Section 5 as Conclusions.

Reviewer 2 Report

Dear editor, thank you very much for your invite. I would like to review this manuscript. This manuscript reported a technique to manufacture Carbon Hybrid Materials (CHMs) 21 by combining CNT non-woven fabric (in the form of sheets or tapes) with microfibers to form CNT-22 CF hybrid materials with new/improved properties. The novelty of this idea is acceptable. The results presented by the author basically support the conclusion. This manuscript can be considered for publication after moderate revisions.

1. “In this work, we will examine the integration of carbon fibers in 38 the intertangled web of CNTs synthesized by the floating catalyst chemical vapor depo-39 sition method in a single-step process. The synthesized carbon nanotube-carbon fiber 40 (CNT-CF) hybrid material can be used to fabricate laminated compo sites with improved 41 properties. In addition, chopped carbon fibers and carbon fiber tows are integrated into 42 the CNT sheets.” These sentences supposed to be placed after the introduction background, the structure of introduction part is suggested to be further optimised.

2. It is recommended to replace Figures with higher resolution for readers to browse.

3. What is crystallographic orientation of CFs-CNTs composites? It is recommended to use WAXD and SAXD for characterization and analysis.

4. It is suggested to supplement the preparation flow chart of CFs CNTs.

Author Response

The responses have been attached 

-------------------------------------------

  1. It is recommended to replace Figures with higher resolution for readers to browse.

Response: Thank you for the comment, the author has provided the editors with all the pictures separately which will be added to the document during the publishing phase for best quality pictures. The picture in the manuscript might have lost some resolution due to uploading and downloading of manuscript on the Nanomaterial manuscript submission website.

  1. What is crystallographic orientation of CFs-CNTs composites? It is recommended to use WAXD and SAXD for characterization and analysis.

Response: We thank the reviewer for the useful suggestions. We are currently working on submitting a beamline proposal for performing WAXD on our samples to SSRL – SLAC and plan on publishing the data in our next paper. We have performed XRD on our CNT samples without any CF discussed in the paper below.

https://doi.org/10.3390/catal12030287

  1. It is suggested to supplement the preparation flow chart of CFs CNTs.

Response: Please see attached document for flow chart

Round 2

Reviewer 1 Report

This manuscript has met the requirements for publication and is recommended for acceptance.

Reviewer 2 Report

I have no suggestions for revision.